# Effect of Soil Creep on the Bearing Characteristics of Soil Slope Reinforced with CFRP and Anti-Slide Piles

Jun Wang [1,*], Lin Liu [1] and Ping Cao [2]

1   School of Building Engineering, Hunan Institute of Engineering, Xiangtan 411104, China
2   School of Resource and Safety, Central South University, Changsha 410083, China
*   Correspondence: 70115@hnie.edu.cn

**Abstract:** In order to research the displacement characteristics and stability of a soil slope reinforced with carbon-fiber-reinforced plastic (CFRP) and anti-slide piles, the displacement composition, aging deformation and failure mode of a soil mass were analyzed. According to the Mohr–Coulomb strength criterion, a new nonlinear, accelerated creep model of soil mass was founded with the addition of a self-building M-C plastic element. Furthermore, a viscoplastic strain analytical formula of an M-C plastic element was obtained, and the tensile deformation characteristics of a CFRP sheet were also discovered under a landslide thrust creep load. According to the environmental conditions of the anti-slide pile, the CFRP was arranged along the load-bearing side of the pile to control deformation. Combining the calculation example, it is shown that the horizontal displacement of the soil slope's composite structure decreases by approximately 40% with CFRP reinforcement. Furthermore, for the first two calculation conditions, after one year, the maximum horizontal displacement decreased by 50% and increased by 10%, respectively. Simultaneously, the overall safety factor increased by 31.3% without soil creep properties. On the contrary, the overall safety factor was reduced, and the slope has a tendency toward unstable failure. Moreover, there is no through plastic zone in the slope. The stability of the reinforced slope and the bearing capacity of the pile are related to the CFRP method. Simultaneously, the structure can reduce the costs and construction difficulty of anti-slide piles in a complex environment surrounded by the soil creep effect.

**Keywords:** CFRP; anti-pile; nonlinear creep; displacement; stability

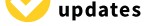



## 1. Introduction

An anti-slide pile is a type of concrete structure which can actively control the deformation of soil, enhance the stability of damaged soil and give full play to the self-bearing characteristics of soil. An anti-slide pile has the foundational characteristics of a large bearing capacity, simple structure and convenient construction. It is widely used in the construction of foundation pits, traffic slopes, mining slopes and other engineering constructions. In general, the anti-slide pile has played an important role in engineering disaster prevention and reduction [1–3]. However, adverse changes in construction conditions and geological and natural environmental factors have an important influence on the bearing stability of the anti-slide pile and can increase the sliding force and displacement of the anti-slide pile's slope structure. They can also decrease the shear strength of the soil and increase the deformation of the pile top and the phenomenon of a sliding body on top. On the other hand, unreasonable design parameters of an anti-slide pile can lead to a reduction in the bearing capacity of the anti-slide pile. Therefore, safety hazards and accidents induced by anti-slide reinforcement are everywhere [4–7]. At the same time, damaged anti-slide piles are reinforced through the following measures, which are often used in engineering: the supplementation of a second source of tension with an anchor cable and enlarging the pile's cross-sectional area, retaining wall, and pile weight before cutting slope to reduce the load. These measures often reinforce the cycle of growth, cost and increasing

construction difficulty. We must find an effective and feasible method. The reinforcement of dangerous anti-slide piles has aroused widespread concern in engineering and academic regions. At present, a large number of soil creep research has focused on linear creep. A two-phase creep element model can only reflect the creep sliding failure of a soil mass. However, the use of a class model and nonlinear creep element model in soil research on anti-slide-pile reinforcement in high and steep slopes is rare, and the slope of sliding soil damage is often caused by accelerated creep. Liu et al. [3] conduced many experiments and established a nonlinear creep model. The analytical solution of the creep model is founded in a combination of the data fitting method and knowledge of elastoplastic mechanics. At the same time, a nonlinear element parameter identification method was proposed. Zhao et al. [8] established a generalized model for rock and soil materials, derived the shear and pressure transfer coefficients through the theory and obtained the coupled evolution equation of the composite fracture creep for rock and soil materials with cracks under complex load conditions and the trend of fracture failure development. Therefore, applying nonlinear creep theory to a geotechnical slope engineering stability analysis is more in line with engineering practice.

Carbon fiber reinforced plastics (CFRPs) are a new type of organic polymer materials, and viscose yarn, polyacrylonitrile fibers and asphalt silk are its basic creations. Furthermore, CFRPs are characterized by their high temperature, high elastic modulus, high strength, corrosion resistance and ease of processing. In particular, in complicated geological conditions, creep displacement and the physical and mechanical properties of the material's stability are presented. It can also prolong the service lives of rock and soil structures [3] as a new and functional material that is widely used in engineering reinforcement. For example, Zhang et al. [9] studied the fracture-bearing characteristics of CFRP-reinforced concrete structures and broadened the application of pile structures in geotechnical engineering. Xiong et al. [10] proposed a bond model of a CFRP and concrete aggregate and revealed the coupling distribution characteristics of the load and stress of the CFRP, concrete and the surrounding rock soil according to the anchoring principle of the soil–anchor structure. However, although there are numerous geotechnical reinforcements methods for anti-slide pile engineering, few consider the instantaneous intensity of soil and the unilateral increase in concrete strength, and the intensity of carbon-fiber-reinforced plastics and their thickness and volume. In addition, the inherent accelerated creep of the soil mechanical properties of an anti-slide pile's load-bearing characteristics is ignored. It is clear that the reinforcement parameters are not reasonable for maintaining the effect on the stability of the slope as a whole. There are even accidents of engineering failure of the anchorage design. The research conclusion and reinforcement effect are limited. The research results show that the nonlinear creep model, the identification of the parameters of the soil pile and CFRP composite structure under disbond and discontinuous deformation conditions, the effect of CFRP yield deformation on crack growth failure, the safety factor and the strength at each point in the composite structure have not been solved, which are bound to be studied further in the future. Therefore, in the present article, three stages of soil pile creep properties are investigated, taking a high and steep anti-slide pile reinforcement and using the soil slope as the subject. The use of CFRP to improve the pile's bearing capacity is investigated, and the deformation of the soil is determined. It can be deduced that this method can control plastic zone development and strengthen the guard slope's stability characteristics for the application of CFRP in infrastructure construction.

## 2. Creep Behavior of Soil

### 2.1. Aging Deformation of Cemented Slope Soil

The reinforcement of a soil slope with an anti-slide pile is mainly used for high and steep slopes formed by artificial activities. In the process of the anti-slide pile's construction and soil excavation, soil stress is released. In fact, the sliding displacement of the adjacent slope often occurs under the action of the dead weight load and pile anchor stress. In the process of sliding deformation, due to the internal friction characteristics of the soil and

the favorable factors retaining the pile structure, the speed of the sliding body gradually slows down, and the slope tends to be stable. However, a retained soil slope that belongs to a permanent structure is often influenced by rainfall, blasting vibrations and artificial disturbances, combined with the properties of the soil creep mechanics that are inherent in sliding displacement. In this case, to reduce the strength of the soil mass, the supported slope structure appears to have a large deformation and displacement mutation. Then, the development of displacement over time will continue. It can be observed that the slope finally shows instability failure. The aging deformation characteristics of the potential sliding body of the slope are shown in Figure 1.

$$u = u_0 + u_c \tag{1}$$

where $u$ is the total displacement of slope soil, m, $u_0$ is the instantaneous displacement, m, and $u_c$ is the creep displacement, m.

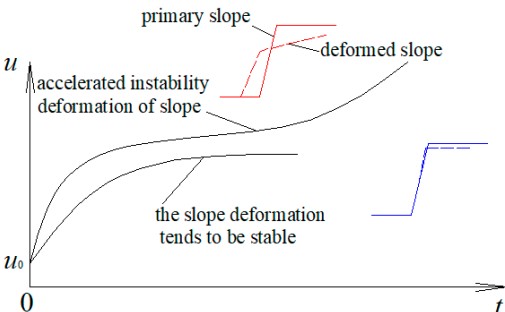

**Figure 1.** Time deformation of slope soil.

At the same time, in the pile reinforcement project, the slope deformation and the state of its stability are analyzed with a lack of real-time tracking. The evolution of the instability of a high and steep soil slope is hard to predict. In order to strengthen the monitoring of the displacement of a slope's soil mass, its aging characteristics were discovered, and a precautionary system of landslide prediction was also established. Improved engineering slope reinforcement is an important topic and a form of disaster prevention.

*2.2. Nonlinear Creep Model*

It is obvious that soil is a type of porous and granular material with low strength. Under the action of an external load, the pore structure and volume of soil are easy to adjust and change. It is easily shown that slippage and the creep phenomenon are caused. When the creep load increases and the shear strength of the soil exceeds the shear strength, the soil presents accelerated creep displacement and damage. Furthermore, experiments show that creep is an inherent mechanical property of soil mass, and a certain amount of creep displacement often occurs in soil under a creep load [10]. When the creep load is less than the soil's shear strength, there are two stages of creep, and the creep is stable. In contrast, the three stages of accelerated creep are not stable. However, a significant amount of soil creep research focuses on linear creep. A two-phase creep element model can only reflect the creep sliding failure of a soil mass. In addition, the class model and nonlinear creep element model are used for high and steep slopes. Research on the anti-slide pile reinforcement of soil slopes is rare, and sliding soil damage to slopes is often caused by accelerated creep. Therefore, a nonlinear creep theory regarding geotechnical slope engineering stability is more in line with engineering practice.

It is known that the difficulty in studying the nonlinear creep of pile-anchored soil is to search for accurate nonlinear elements to reflect plastic deformation. Unlike traditional metal materials, the strength of soil is not only related to its own material properties but also to external stress conditions. The analysis of plastic deformation is more complicated than that of metal materials. For a soil structure reinforced by a concrete anti-slide pile,

the load is mainly carried by the embedded anchorage section. When the soil pressure is high, the accelerated creep displacement effect of the soil has great influence on the internal force distribution of the anchorage section and the plastic zone of the retaining soil structure [11,12]. Therefore, the classical creep element model cannot accurately evaluate the nonlinear bearing capacity of the anti-slide pile. In this paper, according to the displacement characteristics of the soil pile components, the Mohr–Coulomb strength criterion of soil mass was applied. Based on the empirical nonlinear element model theory, the self-built M-C plastic yield element was added to analyze the accelerated creep displacement characteristics of the soil mass. This aspect, named the M-C plastic element, is mainly used to solve the accelerated plastic strain. The entire creep process of the retaining soil was simulated by a parallel connection between the M-C plastic yield element and the three elements of a generalized Kelvin body. Meanwhile, the elastic element was used to simulate the loaded section of the pile, and the viscoplastic element and elastic element were connected in parallel to simulate the anchorage section of the pile. The self-built nonlinear creep model is shown in Figure 2.

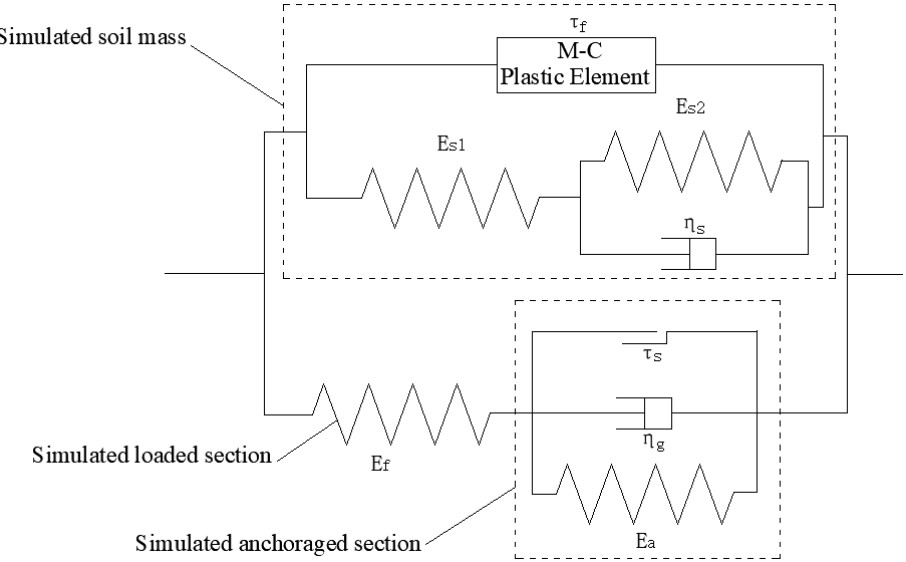

**Figure 2.** Nonlinear creep model of slope reinforcement.

In order to facilitate the nonlinear creep analysis of the loaded section, anchorage section and soil mass, the following assumptions were made:

(1) The soil of the retaining slope is a continuous, homogeneous and isotropic material without groundwater seepage.

(2) The creep load comes from the active and passive earth pressures of the retaining slope soil without earthquake, construction and ground overloading.

Using Figure 2, according to the theory of creep mechanics, the creep synergy equation of a soil pile structure is obtained in Equations (2) and (3).

$$\varepsilon_s(t) = \begin{cases} \sigma_s\left[\frac{t}{\eta_g} + \frac{1}{E_{s1}} + \frac{1}{E_{s2}}\left(1 - e^{-\frac{E_{s2}}{\eta_s}t}\right)\right] & (\tau < \tau_s) \\ \sigma_s\left[\frac{t}{\eta_g} + \frac{1}{E_{s1}} + \frac{1}{E_{s2}}\left(1 - e^{-\frac{E_{s2}}{\eta_s}t}\right)\right] + \varepsilon_{MC} & (\tau \geq \tau_s) \end{cases} \tag{2}$$

$$\sigma_s = \int_0^{L_a} \tau(x)\pi D \mathrm{d}x \tag{3}$$

where $E_{s1}, E_{s2}, \eta_s, \eta_g, \tau_s$ are the elastic modulus, MPa, viscoelastic modulus, MPa, viscoelastic coefficient, MPa·d, viscosity coefficient, MPa·d and the shear yield strength of simulated soil creep, kPa, respectively, $E_f, \sigma'$ simulates the elastic modulus and creep stress of the

pile under a load, MPa, $\tau(x)$ is the shear stress distribution of the soil- pile, kPa $\varepsilon_{MC}$ is the plastic strain reflecting the accelerated creep, and $L_a$ is the length of the loaded section, m.

The plastic strain $\varepsilon_{MC}$ of the nonlinear model can be obtained via the Mohr–Coulomb yield criterion, which is shown in Equation (4).

$$F = -\frac{1}{2}\sigma_s + \frac{1}{2}\sigma_s \sin \varphi + c \cos \varphi \tag{4}$$

According to the conclusion of the creep experiment in Refs. [13,14], the viscoplastic strain rate of the M-C nonlinear creep element can be derived in Equation (5).

$$\dot{\varepsilon_{MC}} = \eta_s \left[ \frac{-\frac{1}{2}\sigma_s + \frac{1}{2}\sigma_s \sin \varphi + c \cos \varphi}{F_0} \right] \left( -\frac{1}{2} + \frac{1}{2} \sin \varphi \right) \tag{5}$$

where $c, \varphi$ are the shear strength index of the soil, kPa and $°$, and $F_0$ is the initial yield strength of the slope soil.

Therefore, the viscoplastic strain of the nonlinear creep element can be obtained via a differential solution. The solution is written as:

$$\varepsilon_{MC} = \int_{t_p}^{t_F} \eta_s \left[ \frac{\frac{1}{2}\sigma_s - \frac{1}{2}\sigma_s \sin \phi - c \cos \phi}{F_0} \right] \left( \frac{1}{2} - \frac{1}{2} \sin \phi \right) dt \tag{6}$$

Combining the plastic yield principle of soil structure, yield failure occurs when accelerated creep is present. Then, the failure time of the accelerated creep can be calculated as follows:

$$t_F = t_0 + \frac{1}{\left[ \int_0^{l/2} \frac{k\varepsilon_0}{d} \exp(\pi d x) - \tau_0 \right]} \tag{7}$$

where $t_0$ is the initial time of the accelerated creep, h, $\tau_0$ is the initial bonding strength of the reinforcement and anchorage agent, kPa, $\varepsilon_0$ is the initial strain of the accelerated creep, $t_P$ is the plastic yield time, h, and $t_F$ is the accelerated failure time, h.

## 3. Load-Bearing Characteristics of Soil Pile Carbon Fiber Reinforcement

### 3.1. Tensile Creep Test of Carbon-Fiber-Reinforced Plastic Sheet

Generally, a CFRP-reinforced anti-slide pile mainly bears a tensile effect. Under the action of landslide thrust, the deformation of the CFRP and pile are coordinated. At the same time, with the increase in the creep deformation of the soil behind the pile, the tensile deformation of the CFRP also increases with time. A carbon fiber epoxy resin composite rectangular sheet was used in the CFRP aging and tensile deformation experiment in this paper. In the following testing parameters, the carbon fiber was a Cymax-L-C type and an XH 130 A/B type epoxy resin adhesive with A:B = 3:1. Regarding the landslide thrust, the tensile load was 400 kN and the creep tensile test time was 45 h. The experimental results of the aging tensile deformation test are shown in Figure 3.

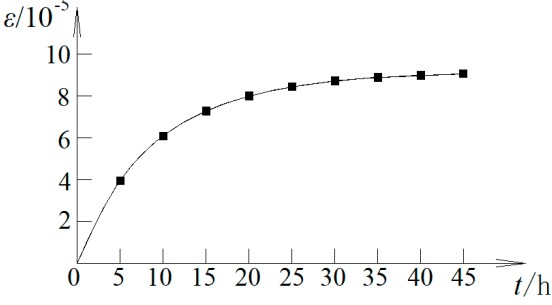

**Figure 3.** Time deformation of sheet material.

It can be seen from Figure 3 that under the creep load, the tensile deformation of the CFRP sheet exhibits stable creep deformation characteristics. In addition, the maximum tensile strain is $8 \times 10^{-5}$, which is smaller than that of the slope soil. On the other hand, the sheet deformation rate attenuated at the initial stage of tensile deformation and remained stable after 20 h.

### 3.2. Load-Bearing Characteristics of Soil Pile–CFRP Composite

Concrete engineering is mainly reinforced with CFRPs for building beam–column structures. Generally, in the form of a reinforcement and sheet, the composite reinforcement structure is formed through external sticking, embedding and concrete. Therefore, according to the pile geometry size, buried conditions, stress characteristics and engineering construction environment, the CFRP placement in the pile is generally under the external load side to enhance the stability of the soil, especially for the reinforcement of the damaged pile.

Under the action of earth pressure, bending deformation and the horizontal displacement of the loaded section of the concrete anti-slide pile occur, which are mainly manifested in the compression of the concrete outside the pile body and the tensile damage of the concrete inside the pile body. Meanwhile, when the load on the top of the loaded section is transferred to the intersection of the loaded section and the anchoring section, the pile body is broken. Therefore, improving the concrete strength of the tensile side of the in-service anti-slide pile or the damaged anti-slide pile has an obvious effect on controlling the further deformation of the soil and pile. CFRP can be closely bonded with concrete via phenolic resin to enhance the tensile strength of the concrete, and it cannot not be corroded by soil and the water in the soil. Therefore, it has a good application prospect for strengthening unstable rock and soil structures. Inside, an anti-slide pile can appear as a composite structure throughout the CFRP reinforcement with concrete, reinforced soil and steel. At the same time, the load-bearing characteristics are above. The strength of the carbon-fiber-reinforced plastic is generally greater than pile strength of the concrete, so the elastic modulus and yield strength of the composite structure are increased. Moreover, the bearing capacity increases and can reduce the creep displacement of the soil mass. Therefore, the stability of the retaining slope is enhanced. A structural diagram of an anti-slide pile reinforced by CFRP is shown in Figure 4.

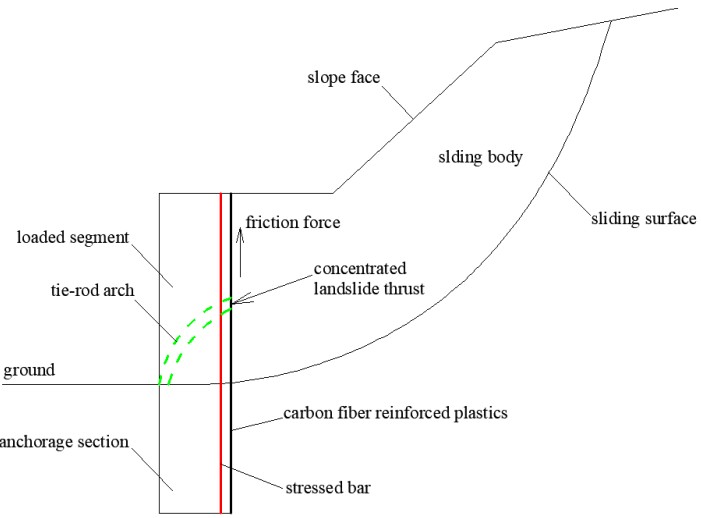

**Figure 4.** Reinforced slope by CFRP anti-pile.

$F_{si}(i = 1, 2, 3)$ is the frictional force, written as:

$$F_{si} = \int_0^A q_{si} dA \tag{8}$$

where $A$ and $q_{si}$ are the frictional area and the frictional intensity, $m^2$ and kPa, respectively.

In order to study the influence of the load section on the sliding force and control deformation better, the shear/span ratio variable is cited in this paper [13,14]. According to the influence of landslide thrust on pile deformation, a tie-rod arch is formed, and then the deformation of the pile under the loaded section is obtained according to the tie-rod arch deformation characteristics. We incorporated the anti-slide pile into the tie-rod arch model (Figure 5):

$$\lambda = \frac{\sum M}{Qh_0} = \frac{(P \cdot a \cdot \cos\alpha + F_{s1} \cdot l + F_{s2} \cdot b - P \cdot b \cdot \sin\alpha)}{(P \cos\alpha + F_{s1}) \cdot h_0} \tag{9}$$

where $\lambda$, $Q$, $P$, $M$, $h_0$, $a$, $l$, $b$ and $\alpha$ are the shear/span ratio, the shear stress, kPa, the concentrated load, kN, the bending moment, kN·m, the effective height, m, the distance between the sliding force and the base, m, the distance between the frictional curve and slide curve, m, the width of the anti-slide pile, m, and the angle determined by the equivalent concentrated thrust and horizontal direction.

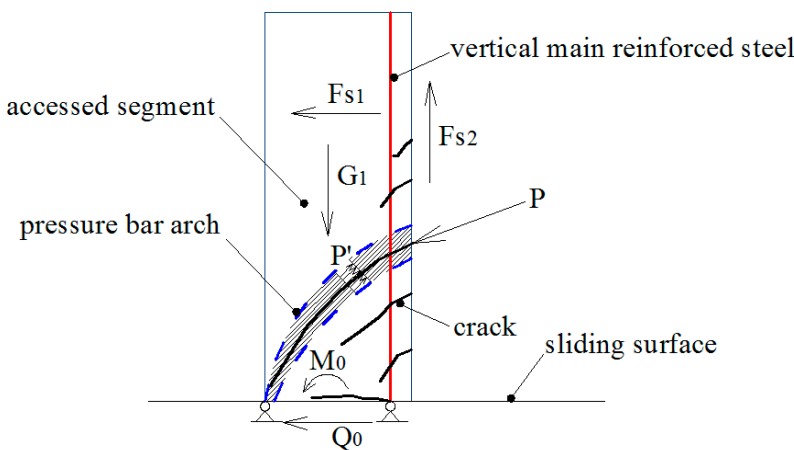

**Figure 5.** Transmission mode and destruction pattern from Ref. [14], copyright 2021, Springer. Q0, M0, Fs1, Fs2, P and Gs1 are the shear force on the sliding surface, the bending moment on the sliding surface, the lateral friction force, the lateral frictional force at the bearing segment, the sliding force, and the gravity load of the bearing segment, respectively.

Equation (9) shows that the shear/span ratio is equal to $\frac{a}{h_0}$ without considering the lateral friction force, $F_{s1}$. Thus, the reasonability of Equation (9) is verified. The thickness distribution of the sliding body and the mechanical monitor results indicate that the distance between the location of the sliding force, P, and the slide surface is about (1/4~1/3)h1. The shear/span ratio ranges from 1 to 5. Thus, shear pressure and inclined tensile failures occur.

When cracks occur in the loaded section, greater deformation and stress redistribution occur. The compression failure occurs in the loaded section. Furthermore, the crack expands further. Finally, the fracture failure of the pile body occurs. There is an obvious correlation between the shear/span ratio and the diagonal dip angle. The crack propagation angle decreases nonlinearly with the increase in the shear/span ratio. This relation is shown in Figures 3 and 6.

The fitting power function equation is:

$$\lambda = m47.017\theta^{-0.8453n} \tag{10}$$

where $m$ and $n$ are experimental constants, and $\lambda$ and $\theta$ are the shear/span ratio and diagonal inclination angle, °, respectively.

Clearly, the shear/span ratio decreases the diagonal dip angle ($\alpha$ in Figure 1). In addition, the decrease rate slightly decreases with the increase in the shear/span ratio. The larger variation in the arch curvature affects the shear strength more significantly.

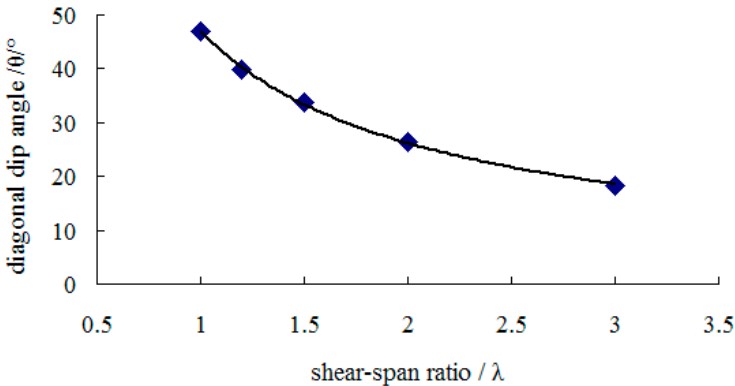

**Figure 6.** Relation between shear/span ratio and diagonal dip angle of arch from Ref. [14], copyright 2021, Springer.

## 4. The Example Analysis

### 4.1. Calculation Range and Calculation Parameters

In order to verify the composite structure system of the CFRP and the anti-pile slope, it should have the advantages of a high efficiency and reasonable ability to control soil deformation, improving the pile strength, reducing the plastic zone and improving safety factor of the retaining side slope. In the present article, a concrete anti-slide pile was selected to reinforce a high, steep homogeneous soil slope. The calculation range of the blocked side slope is as follows: the calculated length is 67 m, the calculated height is 44 m, the slope height is 22 m, and the slope top width is 28 m. The calculated dimensions of the slope height and width meet the requirements of a finite difference numerical simulation calculation. The cohesion is 18 kPa, the internal friction angle is 10°, the bulk density of soil mass is 18 kN/m³, the foundation coefficient of the soil is $3.2 \times 10^4$ kN/m³, the elastic modulus of the soil is 8 MPa, Poisson's ratio is 0.35, the viscoelastic modulus of the soil is 1.25 MPa and the viscoelastic coefficient is 8 MPa·d. The viscosity coefficient of the soil is 18 MPa·d, and the internal force calculation of the anti-slide pile adopts M method, $m = 6 \times 10^3$ kN/m⁴. There is no groundwater, the environment is class I and the safety grade of slope is class I. The size of the anti-slide pile is 2.5 m × 2 m, the length of the pile body is 18 m, the load-bearing section is 8 m, the anchoring section is 10 m, C30 concrete was used for pouring, the vertical stress reinforcement is HRB400 and the side protective layer inside the pile body is 80 mm. Due to the relatively large earth pressure, there is a large bending deformation in the loaded section of the pile. The use of a CFRP sheet to reinforce the inside side of the pile is proposed. The tensile strength of the epoxy resin binder is 30 MPa, the elastic modulus is 3 GPa, the tensile strength of the CFRP is 2 GPa and the elongation is 1.5%. The tangential strength of the bond layer with epoxy resin is 45 MPa. The calculation range of the blocked edge slope is shown in Figure 7, and the grid mesh is shown in Figure 8.

Through the powerful 3D finite difference numerical software, which is based on the Powerstaion platform, the addition of the self-built nonlinear creep constitutive model was applied, which embedded the plastic deformation constitutive model via the horizontal displacement reflection. Using the CFRP composite material model, the soil pile's penal horizontal displacement, the plastic zone and the safety coefficient of the slope were analyzed. In the retaining structure, a brick element was used for soil, a pile element was used for the anti-slide pile, and a shell element was used for carbon-fiber-reinforced plastic sheet. An interface was used for interface bonding. The calculated width was the unit length, and the calculation model had 4896 nodes and 22,334 units. The other boundaries were free. In order to explain the influence of the CFRP and the creep effect on the displacement, the plastic zone and the safety factor distribution characteristics of the composite structure of the retaining soil, a numerical calculation was carried out under the following three conditions: (1) the soil slope was not reinforced with anti-slide piles; (2) the soil slope was reinforced by an anti-slide pile without CFRP; (3) the soil slope was

reinforced by a composite reinforcement of an anti-slide pile and carbon-fiber-reinforced plastic. Non-creep and horizontal creep displacement after one year were considered. The calculation results are shown in Section 4.2.

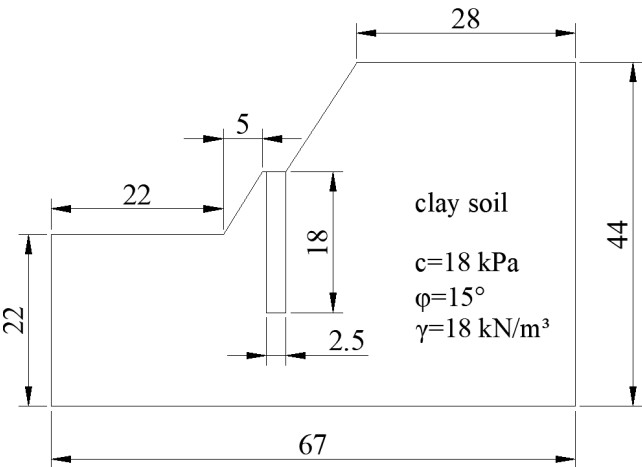

**Figure 7.** Calculation scope of reinforced slope (unit: m).

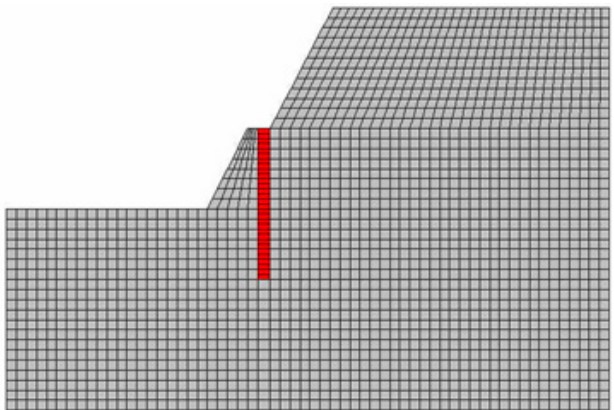

**Figure 8.** The reinforced slope FD calculation mesh.

*4.2. The Calculation Results*

As shown in Figures 9 and 10, under the condition of an unreinforced slope for a numerical calculation of a short duration, the excavation platform was not considered. It is denoted that the horizontal displacement points to the plane negative displacement. The maximal displacement is about 10 cm, and it is mainly distributed in the slope's shoulder position. In the meantime, the minimum displacement is 1.0 cm, and ot is mainly distributed in the local area. Thus, the slope toe produced obvious horizontal displacement and exceeded the standard value. At the same time, a shear plastic zone from the top to the foot of the slope appeared in the shallow part of the slope and extended into the slope. The overall safety factor of the slope is 0.99. Therefore, the slope is in an unsteady state.

As shown in Figure 11, by applying a pile to reinforce the soil slope, the horizontal displacement was obviously improved. Meanwhile, the maximum horizontal displacement is 5.0 cm, which is mainly distributed in the pile top local. The displacement is reduced by nearly 50% when compared to the first case. Therefore, the anti-slide pile reinforcement effect is obvious. However, under the influence of soil displacement creep, the soil pile head is in hidden danger, and is still sliding at the same time as the least-deep horizontal displacement occurs in the slope body figure. Therefore, the deformation tends to be stable.

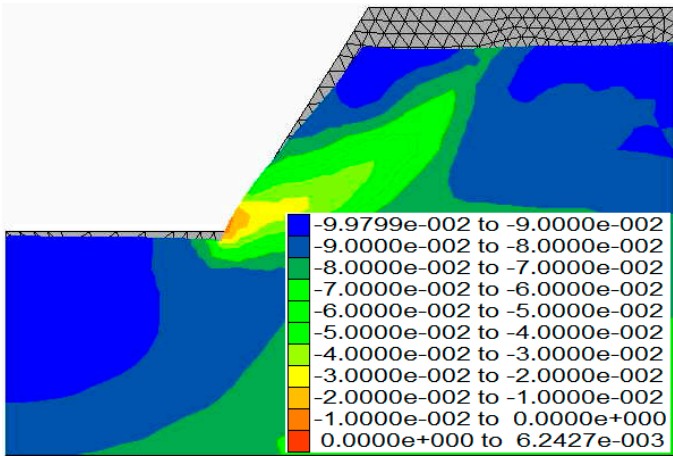

**Figure 9.** Contour of horizontal displacement without reinforcement with anti-slide piles (unit: m).

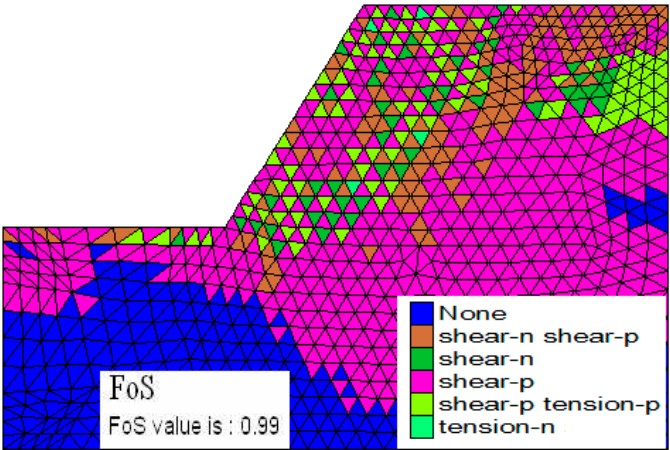

**Figure 10.** Plastic state and safety factor without reinforcement with by anti-slide piles.

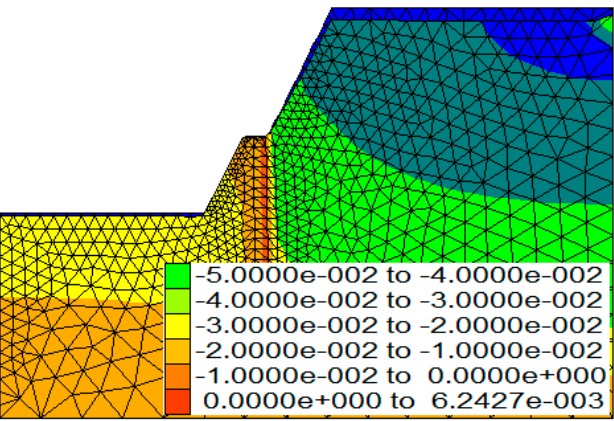

**Figure 11.** Contour of horizontal displacement without CFRP (Unit: m).

In order to further control the creep displacement of the pile top, the pile-bearing capacity was improved to prevent hidden danger. For the CFRP in the side of the pile body, the horizontal displacement was further improved by the block slope (see Figure 12), and the maximum horizontal displacement is 4.0 cm. In addition, the maximum horizontal displacement area shrinks, and the maximum horizontal displacement of the pile-bearing section is 1.8 cm. Meanwhile, its deformation is within the allowed value.

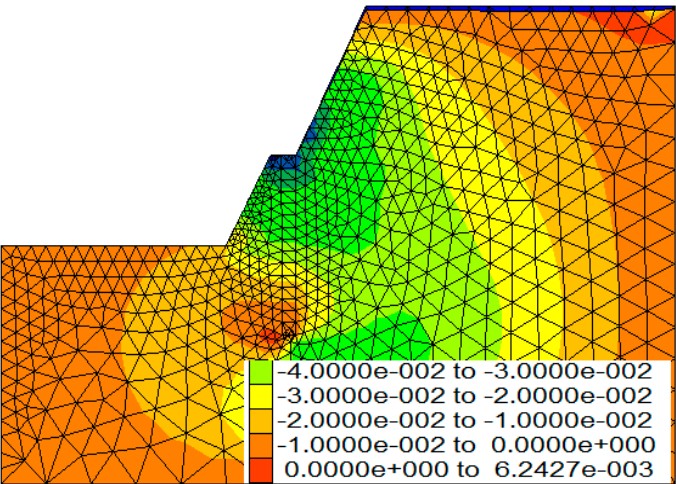

**Figure 12.** Contour of horizontal displacement with anti-slide piles and CFRP (Unit: m).

As shown in Figure 13, concentrating on the shallow surface of the pile's top slope, the plastic zone decreases with CFRP composite reinforcement of the anti-slide pile. There is no transfixable plastic zone appearing in other positions. The overall safety factor of the slope is 1.3. Thus, the slope is in a stable state.

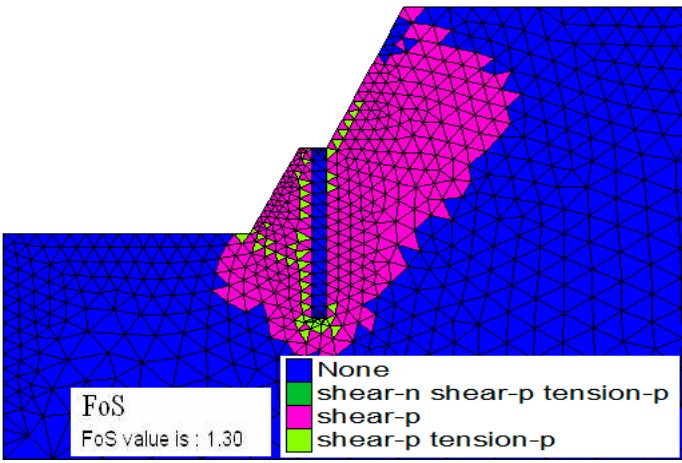

**Figure 13.** Plastic state and safety factor with anti-slide piles and CFRP.

Specifically, the effect of creep on the horizontal displacement of the retaining structure is not considered in the above calculation results. In order to verify the influence of soil creep properties on the bearing capacity of the composite structure system, the horizontal displacement of the soil after one year is obtained by combining the nonlinear creep model in this paper, as shown in Figure 14.

Figure 14 shows that through the pile of the CFRP-composite-reinforced soil slope, the maximum horizontal displacement is about 5.5 cm one year later. Meanwhile, the maximum horizontal displacement of the reinforcement moment increased by 10%, and the maximal displacement and pile top position distribution widened. Thus, the tendency of the increasing displacement leads to a landslide hazard. Therefore, the soil creep effect should not be neglected.

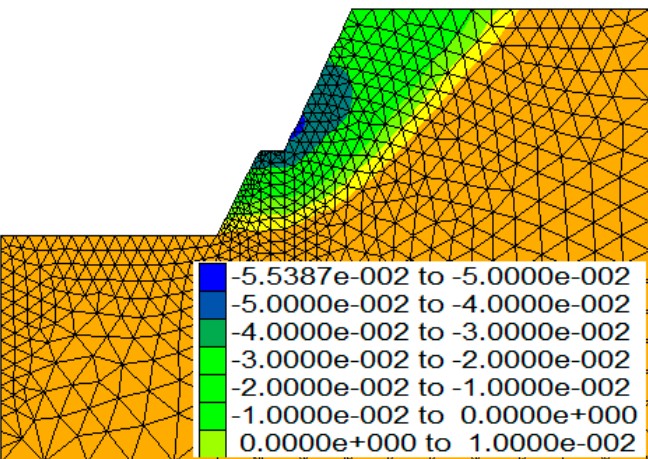

**Figure 14.** Contour of horizontal displacement with anti-slide piles and CFRP under creep (Unit: m).

## 5. Discussion

(1) At present, only the instantaneous strengths of geotechnical materials are considered in the design parameters of geotechnical engineering reinforcements. Based on the external conditions and these parameters, in combination with the design code, the type selection, geometric size and structural design of the reinforcement structure are carried out to ensure the stability of the project. Although the method has summarized many practical experiences and can be applied well to engineering practice, it ignores the inherent creep characteristics of a rock and soil mass and the change in load conditions after stress redistribution. Furthermore, time-dependent deformation, even accelerated deformation, and the weakening of the strength parameters of the composite material, interface debonding and the fracture of the retaining structure and the plastic zone of the perforated rock–soil structure will still occur. Once the deformation exceeds the allowable value, the rock and soil structure will have the possibility of failure due to instability.

(2) Considering the creep effect of soil mass, the slope displacement increases, and the overall safety factor decreases. At the same time, the displacement at the top of the pile increases most significantly. The main reason for this is that the position is located on the slope berm and faces an empty and unsupported structure. Furthermore, the upper slope's mass landslide thrust is larger with the concentration of stress. Thus, the displacement is at the maximum value. However, the displacement at the bottom of the slope is minimal because of the anti-slide piles, which disturb the deformation of the slope. Simultaneously, the limitations of the model presented herein are as follows: ① discontinuous deformation occurs after the debonding of pile–soil interface; ② the pile-soil structure produces accelerated deformation and creep of residual deformation after failure; ③ the soil mass structure changes easily on the heterogeneous macadam soil. The plastic deformation of the anti-slide pile occurs easily when the strength of the loading section is low. In view of these factors, these problems need to be researched further.

## 6. Conclusions

In this work, CFRP is a new kind of reinforcing material for use in a soil slope reinforced by an anti-slide pile. Its application broadens the reinforcement form of concrete structures and promotes the development of modern geotechnical reinforcement technology. In geotechnical reinforcement engineering, it can reduce the dead weight of an anti-sliding structure, improve the bearing capacity and enhance the stability of soil, especially for the reinforcement of an anti-sliding pile structure.

This paper finds advantages in a soil slope reinforced by an anti-slide pile, and a nonlinear creep model which can reflect three stages creep, attenuation creep, stable creep

and accelerated creep, was established. Meanwhile, the creep model and viscoplastic strain analytical formula were found, and the tensile deformation characteristics of the CFRP sheet under landslide thrust creep load were obtained. Combining with numerical examples, it is verified that the reinforcement scheme of laying CFRP on the load-bearing side of the anti-slide pile can significantly control the further deformation of the soil and pile, reduce the penetration of the plastic zone and improve the construction's overall safety factor. It should be noted that the deformation trend of the CFRP composite reinforcement of an anti-slide pile was also discovered while considering the creep effect. In the third case, compared with the first and second cases, after one year, the maximum horizontal displacement decreased by 50% and increased by 10%, respectively. Simultaneously, the overall safety factor increased by 31.3% without soil creep properties. On the contrary, the overall safety factor is reduced, and the slope has a tendency toward unstable failure. Thus, the soil creep effect must be paid attention to in engineering practice.

**Author Contributions:** Data curation, J.W. and P.C.; formal analysis, P.C. and L.L.; project administration, J.W. and P.C.; supervision, J.W. and L.L.; writing original draft, J.W. All authors have read and agreed to the published version of the manuscript.

**Funding:** The research was funded by the Natural Science Foundation of Hunan Province (No.2019JJ40056), scientific Research Foundation of Hunan Province Education Department (No.18A345, 18B391, 22C0765),and the construct program of applied specialty disciplines in Hunan province.

**Institutional Review Board Statement:** Not applicable.

**Informed Consent Statement:** Not applicable.

**Data Availability Statement:** All data used in this paper are listed in the tables and figures of this manuscript.

**Acknowledgments:** The authors gratefully acknowledge the support from the Natural Science Foundation of Hunan Province (No. 2019JJ40056), the National Natural Science Foundation of China (No.51804110), the Scientific Research Foundation of Hunan Province Education Department (No.18A345, 18B391, 19B124), and the construction program of applied specialty disciplines in Hunan province. These works are gratefully acknowledged.

**Conflicts of Interest:** The authors declare no conflict of interest.

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
