# Peer review of "Effect of Soil Creep on the Bearing Characteristics of Soil Slope Reinforced with CFRP and Anti-Slide Piles"

_coatings, doi:10.3390/coatings13061025_

Round 1
Reviewer 1 Report
please see the attachment

Author Response
Dear Expert:
Thanks for your hard work and kind recommendation on our manuscript. According to your comments, we have revised our manuscript. Please see the attachment.
Best wishes to you.
Sincerely Yours,
Authors of this manuscript

Reviewer 2 Report
GENERAL COMMENTS
The work entitled “Effect soil creep on bearing characteristics of CFRP and antislide piles reinforced soil slope”
RELEVANCE (considering the contribution to the advancement of knowledge): Good.
ORIGINALITY (considering the problem to be studied and the existing knowledge gaps that justify the study): Good.
TECHNICAL AND SCIENTIFIC MERIT: Good.
FINAL OPINION: The work has potential and merit to be published.

The English at work is very good.
Author Response

(The authors gave the same response as above.)

Reviewer 3 Report
Dear Authors,
Your manuscript is a valuable scrupulous multifaceted well-illustrated and based on a mathematical modelling contribution with a high scientific output. You have used up to date methods. The study of anti-slide soil slope reinforcement is an important issue.
I propose you to give some explanations and make a few corrections. The details are presented herewith below.
Major comments
Line 99
2.2. Nonlinear creep mode
Line 172
3.1. Tensile creep test of carbon fiber reinforced plastic sheet
Where is the title of your section 3?
Figures 8-14
The figures are very interesting.
There is no information about vertical and horizontal scale of the figures. If they are the same, then the slopes are very steep. It is unlikely that the soil on such a slope will be stable. If waterlogging is added to this, then a landslide is inevitable. According to the Figure 14, the soil mass to the right of the pile is greater than that to the left of the pile. The mass of the landslide will be able to shift to the left both the pile and the soil located on the slope below the pile.
Minor comments
Line 245
Stress redistributions occur in the beam when cracks at the loaded segment form.
An important issue of the constructs related to the natural soil and different building soils are the water regime deformation, slope design and building soil stabilization by drainage. Have you accounted these aspects?
Line 265-267
…to reinforce a homogeneous high steep soil slope.
…the calculated length is 67 m, the calculated height is 44 m, the slope height is 22 m, and the slope top width is 30 m
The slopes are too steep. Such slopes can be stabilized using more berms and introducing more gentle slope into the project.
Lines 301-303
…the soil slope was reinforced by anti-slide pile without carbon fiber reinforced plastics; (3) the soil slope was reinforced by composite reinforcement of anti-slide pile and carbon fiber reinforced plastic…
Please compare both anti-slide piles in Conclusions.
Author Response

(The authors gave the same response as above.)

Round 2
Reviewer 1 Report
All major comments were adequately addressed and the Authors have done an admirable job of improving the quality of the manuscript. Therefore, it can be accepted without any structural modification.